# The Perceptions and Use of Urban Neighborhood Parks Since the Outbreak of COVID-19: A Case Study in South Korea

**DOI:** 10.3390/ijerph20054259

**Published:** 2023-02-27

**Authors:** Jiku Lee, Jinhyung Chon, Yujin Park, Junga Lee

**Affiliations:** 1Department of Environmental Science and Ecological Engineering, Korea University, Seoul 02841, Republic of Korea; 2OJEong Resilience Institute, Korea University, Seoul 02841, Republic of Korea; 3Division of Environmental Science & Ecological Engineering, Korea University, Seoul 02841, Republic of Korea; 4Department of Forestry and Landscape Architecture, Konkuk University, Seoul 05029, Republic of Korea

**Keywords:** COVID-19, stress, urban parks, motivation, causal loop diagram

## Abstract

As the COVID-19 pandemic continues, the stress of city dwellers is increasing, and some adapt to the pandemic by pursuing physical and psychological well-being in neighborhood parks. To improve the resilience of the social-ecological system against COVID-19, it is important to understand the mechanism of adaptation by examining the perception and use of neighborhood parks. The purpose of this study is to investigate users’ perceptions and use of urban neighborhood parks since the outbreak of COVID-19 in South Korea using systems thinking. To verify the hypotheses about the relationship between variables involved in COVID-19 adaptive feedback, two research objectives were set. First, this study determined the causal structure leading to park visits using systems thinking. Second, the relationship between stress, motivation, and the frequency of visits to neighborhood parks was empirically verified. To conduct the research, the system of use and perceptions of parks were analyzed through a causal loop diagram to determine the feedback between psychological variables. Then, a survey was conducted to verify the relationship between stress, motivation for visits, and visit frequency, which are the major variables derived from the causal structure. A total of three feedback loops were derived in the first step, including a loop in which COVID-19 stress was relieved by visits to parks and a loop in which COVID-19 stress worsened due to crowding in parks. Finally, the relationship of stress leading to park visits was confirmed, and the empirical analysis showed that anger about contagion and social disconnection were linked as motives for park visits, and that park visits were mainly motivated by the desire to go out. The neighborhood park functions as an adaptive space for the stress of COVID-19 and will maintain its role as social distancing becomes more important to various socio-ecological changes. The strategies driven by the pandemic can be adapted in park planning to recover from stress and improve resilience.

## 1. Introduction

COVID-19 has accumulated many positive cases worldwide since its outbreak in 2019, and its impact has continued for over two years. Global trends indicate that the pandemic seems to be gradually declining [1]. In the case of South Korea, however, the virus repeats a trend of lulls and activations, and South Korea is experiencing its seventh epidemic in 2022 [2]. As the pandemic continues, social capital decreases, new variants of the virus emerge, and the stability of the social-ecological system is further threatened [3].

COVID-19 has caused many changes in daily life due to its high contagiousness. The virus affects the whole of society and causes problems in areas such as nutrition, lifestyle, education, economics, and quality of life [4,5]. In addition, it was found that the COVID-19 lockdown contributes to brain hemodynamics, causing psychological disorders such as depression, anxiety, and insomnia [6,7]. South Korea positively enacted an initial response and quarantine policy against COVID-19, especially in its governance, medical system, entry prohibitions, self-quarantine, and contact tracing [8]. In this process, however, individual isolation has occurred [9]. Isolation from networks may have led to effective quarantine, but it caused individual and societal depression, known as the Corona blues [10]. As people refrain from going out and spend more time indoors, they feel isolated, concerned about the news of COVID-19 as reported in the media, and anxious about becoming infected. This increases various psychological stresses, such as anxiety, depression, fear, anger, and loneliness [11]. A total of 36.8% of South Koreans have experienced or been diagnosed with depression, ranking the highest among OECD nations [12]. As the impact of COVID-19 continues, damage to health and disruptions to daily life are increasing.

As the virus response system has been stabilized in South Korea, there are no special social distancing requirements except for the quarantine of COVID-positive patients. It has been 3 years since the outbreak of COVID-19 and South Korea is still experiencing several waves, even after the regime was shifted to the “living with Corona” strategy. In this context, distancing will still be important in the living with COVID system. The Corona blues is directly related to social distancing, because loneliness arises from isolation. However, in addition to the loneliness, depression also causes health concerns, the breakdown of social relationships, economic difficulties, etc. In South Korea in particular, social stigma as an infected person and self-criticism for infecting those around you were significant causes of negative emotions [13]. Therefore, regardless of the specific phenomenon, it is necessary to pay attention to the Corona blues as well as its treatment and recovery from a long-term perspective.

COVID-19, which is believed to have originated from wildlife, has a significant impact on people’s health and well-being. Since COVID-19 affects people’s physical and mental health, it highlights the connection between human health and the natural environment. Therefore, it is crucial to prioritize and manage the social and ecological ecosystems in our living environment. A neighborhood park can be a space that performs such a function in the daily environment. Among the changes that have emerged in the COVID era, due to increased interest in psychological health, one of the most notable is that people started spending more time in green spaces, especially in parks close to their homes [14].

Neighborhood parks have the advantage of high accessibility because they are located within neighborhoods, so they function as an alternative leisure space and induce people’s visits [15]. It was found that the more severe social regulations are, the greater the protective effect of easily accessible green space [16]. Parks where people can connect with nature improve people’s psychological well-being by regulating personal stress and providing them with social and cultural services. This is a highly prized function, especially in green spaces close to residential areas [6,17,18,19,20]. In addition, it was found that people visit parks to have an alternative space in which to exercise for health purposes. As such, parks and green spaces, which provide a space to interact with nature, are acting as adaptive spaces for physical and mental health in the COVID era over both the short and the long term and improving social-ecological resilience by increasing the adaptability of individuals and of society to COVID-19 [21,22].

To recover the resilience of the social-ecological system, rather than just understanding the phenomenon, it is necessary to identify the structure of the phenomenon according to the flow of the system based on the time sequence. It has been revealed that urban parks exert an important impact on physical and mental well-being and that users give positive meaning to green spaces during a pandemic [23,24]. However, prior research on the relationship between COVID-19 and parks has tended to focus on analyzing the amount of physical movement in parks or the frequency of visiting parks according to personal and social characteristics. This study aims to investigate how the relationship between the perceptions and use of urban neighborhood parks is affected by the COVID-19 outbreak in South Korea from an integrated and systematic point of view. This study has two main objectives: first, to determine the causal structure of users’ perceptions and behaviors regarding neighborhood parks, and second, to examine the process of users’ cognition, perception, and behavior based on the derived causal structure. The following hypotheses aim to achieve the objectives.

**Hypothesis 1 (H1).** 
*Various COVID-19 stresses increase motivation to visit parks.*


**Hypothesis 2 (H2).** 
*The motivation to park visits is significantly linked to actual park visits.*


The framework of the study is shown in Figure 1. This study not only explains the phenomenon of increased visits to neighborhood parks, but also reveals the reason why people visit during the COVID era. This can contribute to establishing a framework for land use planning, allocation, and park management by reflecting the needs of people regarding parks in the post-COVID era.

## 2. Materials and Methods

### 2.1. Causal Structure of the Perception and Use of Urban Neighborhood Parks

A causal loop diagram (CLD) is a method to understand the causal structure of variables from the perspective of systems thinking. Systems thinking is used to understand the complex and dynamic structure of a system based on the causal relationship between various variables that organize the system [25]. In other words, a CLD is a tool that expresses the dynamic and interconnected conditions of the elements constituting a system [26]. Based on the dynamics of a system, the CLD shows the dynamic aspects of that system over time, which allows for an analysis of structure and pattern over time. However, since a social-ecological system is intricately and nonlinearly composed, the boundary of the system is generally unclear and hard to define. The pattern and structure of a system can be clearly apprehended by setting the spatiotemporal boundaries of the system. The existing system of parks and users has changed due to the disturbance of COVID-19. Therefore, in this study, the system is defined as the perception and use of park users, and its spatiotemporal boundaries comprise each neighborhood park and the period since the outbreak of COVID-19. Since South Korea has been experiencing infection waves since 2019, the number of COVID positives are regularly maintained in the tens of thousands. Thus, the external conditions other than the behaviors in the parks was assumed to be similar to the early stage of the virus outbreak. The complex system of people’s behavior in the context of COVID-19 can be explained using the concept of social-ecological systems, so it is appropriate to use systems thinking.

#### 2.1.1. Identifying Relevant Elements

During the COVID-19 outbreak, the scope of the change in behavior and the perception of parks by users became issues of interest. Since different perceptions of and purposes for visits to parks came into play after the advent of COVID-19, this study focused on descriptions of actual users’ experiences or first-hand observations. To map out the changes in the use of parks following the outbreak of COVID-19, we collected articles and reports about the current state of park visits. Although this study specifically focuses on outdoor space visitation patterns and user perceptions and attitudes in South Korea, it is intended to provide insight into the general structure of park visitation systems by drawing from the broader literature on environmental psychology. The Science Direct and Scopus academic databases were used to derive variables from prior research. Among the articles from 2020 to 2022, after the outbreak of COVID-19, journals such as UFUG (Urban Forestry and Urban Greening), IJERPH (International Journal of Environmental Research and Public Health), Frontiers in Public Health, and Journal of Environmental Management and Tourism were reviewed based on keywords related to changes in the frequency of visits and attitudes, roles of green spaces, etc. This includes examining behavior patterns following epidemics or disaster, exploring the human perception of green spaces, and reviewing studies on how biophilia and general motivation for park visitation responded to COVID-19 issues. However, studies targeting specific age and social groups have been excluded. Interviews and blogs were also examined to investigate real experiences related to neighborhood parks. The visitors’ tendency to seek alternative spaces for indoor spaces was identified, with the emotions they felt about visiting the park. Data about people’s stress due to the pandemic were collected through research papers that used surveys. For this, stress in disaster, crisis, and pandemic situations was identified based on the existing stress rating scale. Moreover, the response of parks to COVID-19 was collected by referring to landscape articles from ASLA (American Society of Landscape Architects), Lafent (Green Culture and Arts Portal of Korea), and other landscape architect magazines. Through this, the concepts and major variables surrounding the relationship between COVID-19 and park use were derived.

#### 2.1.2. Determining Relationships

Based on the variables derived in the preceding step, we determined the main variables that play a central role and identified the causal relationship between these variables. It is important to distinguish the polarity of the relations according to the direction of the response between variables.

#### 2.1.3. Forming Feedback Loops

A causal loop diagram was created using the variables and causal relationships of the relation between park use and COVID-19. Several causal relationships came together to form feedback, which is a closed sequence of causes and effects [26], and this feedback includes a reinforcing loop and a balancing loop. In a reinforcing loop, variables respond in the same direction if they are in a positive relationship causing accumulated effects, whereas in the balancing loop, one variable responds in the opposite direction to the other variable, and the effect is finally offset in the feedback. The Vensim PLE x64 program (Ventana Systems, Harvard, MA, USA), a system dynamics modeling program, was used to make the causal loop diagram.

#### 2.1.4. Analysis of Feedback Loops

Finally, the feedback structure from the CLD was analyzed. At this point, a control variable that can change the direction of feedback could be derived, and the problem could be solved by controlling this strategic point.

### 2.2. Causal Relationship between COVID-19 Stress, Motivation, and Frequency of Visits

Based on the causal map, a survey was conducted to empirically verify the detailed relationship between park users’ cognition, perception, and behavior in South Korea. The variables used for examination were the 3 main concepts of COVID-19 stress, motivation for visiting neighborhood parks, and frequency of park visits, which were derived from the causal structure detailed in the preceding steps.

#### 2.2.1. Participants

The survey was conducted with people who had visited the park both before and after the outbreak of COVID-19. A random sample of people aged 10 to 60 was asked to answer a survey about their experience with the park during the COVID era, and a total of 114 responses were submitted. Participants were asked their age, biological sex, and whether they had traveled to neighborhood parks by vehicle or on foot, and the scope of the neighborhood was set autonomically for each respondent. Responses that did not have any visits to neighborhood parks either before or after the outbreak of COVID-19 were excluded from the analysis, as they were judged not suitable for the sample.

#### 2.2.2. Questionnaire

Since stress manifests in various ways, a stress evaluation index must be prepared for each specific situation. As such, an evaluation tool for COVID-19 stress is being developed, and in this study, questionnaires were constructed based on that tool. The COVID-19 Fear Scale (FCV-19 Scale) has been used since it was first presented as the first standardized scale to address psychological stress caused by COVID-19 [27]. Additionally, the CSS (COVID Stress Scale), which examines COVID-19 stress from various angles dealing with fear, trauma, socioeconomic cost, and so on, was used to modify the context [28]. The COVID-19 Stress Scale for Korean People (CSSK) was used without modification; it is an index that reflects South Korea’s unique quarantine policy, culture, and values [29]. Furthermore, ref. [30] was referred to for the composition of 19 items used in the survey. The scale and its references are organized in Table 1.

#### 2.2.3. Procedures

For ease of access to the survey and to address the difficulty of face-to-face surveys caused by COVID-19, the survey was produced in Google Forms and distributed online (Appendix A). The survey was distributed over a period of 5 days from 8 December to 12 December 2021. Respondents were not compensated for the survey, randomly collected, and the confidentiality of responses was guaranteed.

#### 2.2.4. Analysis

Of the 114 responses collected, only 98 were judged as valid responses and were used for the empirical analysis. A total of 98 valid questionnaires were used for the empirical analysis. The IBM SPSS Statistics 26 statistical program was used for data analysis (SPSS Inc., Chicago, IL, USA). The variables of COVID-19 stress and motivation to visit parks were classified through principal component analysis. Principal component analysis with varimax rotation was performed on each of 19 items for both the COVID-19 stress and motivation variables, and factors and factor scores were derived. Then, correlation analysis was performed based on the factors extracted in the preceding step to analyze the causal relationship between COVID-19 stress and park visit motivation. Since the direction and flow of users’ perception and behavior have already been fixed in the causal loop diagram, analyzing the tendency and relationship was thought to be sufficient. Thus, only Pearson correlation analysis was conducted rather than a complex method. Moreover, the sample is somewhat skewed, so it was believed that the reliability would be low when the regression analysis was conducted. Finally, the association between the frequency of visits and visit motivation was also examined through correlation analysis.

## 3. Results

### 3.1. Causal Structure around Neighborhood Parks Based on the COVID-19 Response

The feedback structure of increasing resilience against COVID-19 by visiting a neighborhood park was expressed as a causal loop diagram (Figure 2, Table 2). As COVID-19 spreads, restrictions in daily life occur, and concerns about infection increase, which increases individuals’ COVID-19 stress [38] (Table 3). This leads people to visit neighborhood parks for stress relief. Through physical activity in the park, health is improved, which then induces a causal structure that reduces the number of COVID-19 infections. On the other hand, as users become concentrated in neighborhood parks, the risk of COVID-19 infection increases, which may increase COVID-19 stress. In addition, neighborhood parks can strengthen their resilience by changing their physical structure and diversifying support services. Introducing special designs such as social distancing designs and linear structure in parks, reducing the risk of infection by minimizing face-to face contact and reflecting the principle of social distancing have the potential to improve society’s adaptability to disturbances. Parks can also relieve individual stress by improving services in support of a variety of activities within the park. As such, neighborhood parks can function as resilient spaces, and their resilience can be improved through physical improvement and service expansion [30].

Looking at the causal structure above, it can be seen that feedback is ultimately formed around three main variables (Figure 3). Stress is caused by the difficulty of engaging in outside activities, a goal is formed to relieve this stress, and a visit to the neighborhood park is finally made. On this basis, this study constructed a survey to empirically verify the causal relationship by focusing on these three variables. It was confirmed that the causal structure follows the general process of environmental psychology in which cognition leads to perception and then behavior.

### 3.2. Causal Relationship between COVID-19 Stress, the Motivation to Alleviate It, and Visits to Neighborhood Parks

#### 3.2.1. General Characteristics of Respondents

A survey was conducted for empirical verification based on the causal map derived in the preceding step. Although the number of valid responses, 98, is somewhat insufficient, it was judged to be sufficient for the analysis since it was more than the minimum number of samples of 30 with a normal distribution. The general characteristics of the respondents are shown in Table 4. Regarding the sex of respondents, women (76.5%) were more than three times more prevalent than men, and the age distribution was highest among those in their 20s (50.0%) and lowest among those in their 10s (6.1%) and 60s (3.1%). More than 90% of participants visited neighborhood parks within a 30 min radius from their homes on foot. In terms of accessibility, it was judged that the participants’ environments were all similar.

#### 3.2.2. Factor Analysis of COVID-19 Stress and Motivation for Visiting Neighborhood Parks

In this study, the factors explaining two variables selected for empirical verification were extracted through factor analysis: COVID-19 stress and motivation to visit neighborhood parks. First, 19 items measuring COVID-19 stress were used in the principal component analysis. Five factors with eigenvalues higher than 1.0 were derived through varimax rotation for the simplification of factor loading values. The KMO value was 0.819, confirming that the sample was suitable for principal component analysis. Each factor was named to reflect the representativeness of its constituent items: anxiety about infection, helplessness due to social disconnection, anger over contagion, traumatic reaction, and daily stress (Table 5). Cronbach’s α values for each factor were all above 0.7, confirming that the satisfactory consistency of the measured values was satisfied [44].

Principal component analysis using varimax rotation was also performed on 19 items for the motivation to visit neighborhood parks. The commonality and the MSA were found to be higher than 0.5, and the KMO test had a value of 0.765. It tests whether the sample results are suitably predicted by the factors, and it is judged to be enough when the value is more than 0.7. Six factors with eigenvalues higher than 1.0 were derived (Table 6); eigenvalues refer to the sum of the variance that can be explained by a factor, and refer to the extent of how much a factor can explain each variable. According to the characteristics of the constituent items, the names of the individual factors were “biophilia,” “exercise for health improvement,” “social and leisure activities,” “safety from infection in the park,” “meeting the desire to go out,” and “alternative place exploration.” Cronbach’s α value for each factor was 0.6 or higher, confirming that the reliability was acceptable [45].

#### 3.2.3. Effect of COVID-19 Stress on the Motivation to Visit Neighborhood Parks

Correlation analysis was performed using the type of COVID-19 stress as the independent variable and the type of motivation as the dependent variable to understand the causal relationship between COVID-19 stress and the motivation to visit neighborhood parks (Figure 4). The analysis showed that the “helplessness due to social disconnection” factor was linked to “meeting the desire to go out” but not to “social and leisure activities” and “safety from infection in the park” factors. This can be interpreted as people choosing a neighborhood park as a good place to go out to even though they do not perceive the park as safe from COVID-19 infection. These linkages can also be seen as people visiting a neighborhood park to achieve the purpose of going out rather than to engage in various activities in the park, which means that the key role of the park in the COVID era is to leave the house. It can be inferred that this is because contact and interaction with nature can relieve stress and encourage psychological stability [46,47]. On the other hand, the “traumatic reaction” factor had a positive correlation with “social and leisure activities,” meaning that people gained relief from the physical traumatic reaction to COVID-19 through social and leisure activities and that the park provides a space for improving community resilience.

According to the results, Hypothesis 1 is partially accepted. Some of the stress variables had a negative link with motivation; in particular, COVID-19 stress occurred mainly due to “helplessness due to social disconnection,” and other stress factors were not connected to positive motivation to visit the park.

#### 3.2.4. Effect of Neighborhood Park Visit Motivation on Visit Frequency

To verify the correspondence between the frequency of visits to neighborhood parks and the motivation for visiting neighborhood parks, a correlation analysis was performed with the group of increased frequencies of visits and that of high frequencies of visits. As a result, the factors “exercise for health improvement” (*p* = 0.045) and “meeting the desire to go out” (*p* = 0.000) showed a correlation with the increased frequency group, and “biophilia” (*p* = 0.031) and “meeting the desire to go out” (*p* = 0.012) showed a positive correlation with the high frequency group (Figure 4). The reason for the lack of a significant influencing relationship between biophilia, the first factor among the motivations for visiting neighborhood parks, and all stress factors may be because biophilia is the general motivation for park visits regardless of COVID-19; thus, there does not appear to be a significant difference in the motivation for park visits before and after the disease outbreak [48,49]. The motivation of desire to go out significantly affected both target frequency groups. Due to the strengthened social distancing regulation, the pattern of visiting neighborhood parks appears to be a common method of overcoming the limits on outdoor activities. On the other hand, since visits to parks to improve health displayed a significant correlation with only the increasing group, people became more aware of their health after the outbreak of COVID-19, and they visited parks as an alternate exercise space. This is consistent with the findings of existing research that visits to accessible parks improve health and enhance adaptive capacity against COVID-19 [47].

The hypothesis about the relationship between the motivation to visit parks and the frequency of visits was also partially accepted. Not all variables of motivation were linked to actual visits to neighborhood parks, but because several motivation factors were connected to increased visits, the hypothesis can be partially accepted. Among the motivation factors, “meeting the desire to go out” was the only factor that was linked from the stress factor to an increase in visit frequency, which contributed to the support for Hypothesis 2.

## 4. Discussion

In general, the system of external stimuli leading to human response undertakes a process of sensation to cognition, perception, value judgment, and attitude. Through this study, it was found that there is a process in which stress and then motivation to visit a park occur, after receiving an external stimulus based on COVID-19, and before leading to the behavior of visiting a park.

### 4.1. COVID-19 Stress and Motivation to Visit Parks

Among the types of COVID-19 stress, “helplessness due to social disconnection” and “anger over contagion” had a negative effect on “social and leisure” activities and “safety of infection in the park.” These two motivation factors for visiting neighborhood parks were not all related to the actual visit, while “meeting the desire to go out” was the only motivation factor connecting stress and actual visits. On the other hand, “anxiety about infection” and “helplessness due to social disconnection” appear to be the main factors of factor analysis for COVID-19. Since “anxiety about infection” is a factor related directly to health, the “helplessness due to social disconnection” factor is the most dominant psychological stress caused by COVID-19. This large flow indicates that the neighborhood park is just a place to go out to rather than a place in which to engage in specific activities. At the same time, being isolated indoors without being able to go outdoors freely due to social distancing causes the greatest stress in the COVID-19 pandemic.

### 4.2. Park Visits as an Adaptive Behavior

The causal structure around neighborhood park visits following the outbreak of COVID-19 shows the capacity of improving the resilience of individuals and society by park visits relieving pandemic stress. Feedback was formed between stress, park visit motivation, and visit frequency. More specifically, a causal relationship was established in which a sense of isolation due to social disconnection was extended to the willingness to go out, which led to an increase in visits to the park. People’s purpose was that of visiting the park itself, rather than visiting the park to engage in other activities or in social interactions. When an external stimulus (COVID-19) occurs, the individual and the environment interact based on affordance in the process of accepting the stimulus [50]. As a result, an individual obtains a negative attitude as a response to stress and an adaptive behavior to relieve anxiety by identifying attitude with behavior; that is, they engage in a behavior to relieve helplessness due to social disconnection, which manifests in the form of a visit to the park. In other words, residents are motivated to visit neighborhood parks because they know that they can recover from stress, improve their physical well-being through exercise, and satisfy their individual needs by visiting neighborhood parks. The park was simply an alternative place to go out during the pandemic, and it can also be said that it works as a refuge and a space of resilience.

South Korean academics have argued for the expansion of park infrastructure to accommodate the increase in park visitors since the outbreak of COVID-19 [51,52,53]. However, the Ministry of Environment has not promoted relevant projects, and local governments have not presented any plan to expand the parks with a focus on their recovery function. South Korea has focused solely on short-term disease management in response to COVID-19, rather than developing a resilient recovery strategy. This approach is contrary to the findings of this study, and it is essential to expand parks from a long-term perspective to provide restorative effects, improve accessibility, and expand the user space by further developing park infrastructures.

As an alternative to restoring social-ecological resilience against COVID-19, infection-free park design can mitigate the conditions of the pandemic. For example, Domino Park in the United States supported park activities during COVID-19 while encouraging the maintenance of social distancing by arranging circles with a diameter of 2.4 m on its lawn [54]. Additionally, many other parks converted trails into one-way trails to help people minimize contact with others [55,56]. “The Invisible Facemask,” which won the Social Architecture Post-COVID General Idea Contest, is a design that minimizes contact by arranging vertical intersections and pocket spaces [54]. These design strategies minimize face-to-face contact and lower the risk of infection, ultimately contributing to the overall resilience of the system. This creates a process in which individuals and social-ecological systems adapt to COVID-19, and ultimately, resilience improves.

Since the sample used for the empirical verification of this study was insufficient, it can be suggested that COVID-19 stress and the major motivational factors for visiting parks are not organically linked to park visits. Additionally, as the sample is skewed, it may not include the general experiences and perceptions of the population. In addition, there are more factors that influence park visits, such as the meaning and emotion of neighborhood parks or the spread of COVID-19 [57]. Therefore, additional considerations of behavior in parks, the environment of neighborhood parks, and the environmental conditions that affect individual park visits are needed. Additionally, regarding the motivation to visit parks, only the factors related to psychological stress were used for analysis in this paper; therefore, by addressing the other sociocultural and physical factors affecting this motivation, such as park facilities and programs, a more holistic outcome for the park can be achieved [57,58].

## 5. Conclusions

This study was conducted with a focus on the behavior and perception of park users to verify the causal structure of neighborhood parks. In the relationship between COVID-19 and parks, prior research has focused on identifying phenomena such as the increase in the percentage of people visiting parks. This study extended previous work by further investigating the causal relationship between pandemic stress, motivation and the frequency of park visits from a systematic perspective. As indicated by the causal loop diagram, the system of the neighborhood park consists of balancing feedback that leads to the recovery of COVID-19 stress and a decrease in the number of positive COVID-19 cases as well as reinforcing feedback that increases the risk of outdoor infection. A system involving park users’ stress and their motivation to visit was designed from the perspective of adaptation to the disease environment. Empirical verification was conducted based on this finding. The intention to visit neighborhood parks mainly stems from the helplessness caused by restrictions on social interaction and going out, anger about the possibility of contagion from others, and the physical reaction caused by COVID-19 stress. The motivation to enjoy nature in the park and relieve helplessness by going out has a significant effect on visits to parks.

As the social-ecological system has been threatened by numerous disturbances, some spaces have served as buffers from these disturbances, while some spaces have gathered people and created communities, ultimately improving the resilience of the system. The social-ecological system will be severely threatened by more dynamic disturbances in the future, and recovering the system in high-adaptability spaces will be the most important task when a problem occurs. Some spaces have promoted the recovery of people and society simply by existing, such as neighborhood parks in the COVID-19 era. The results show that rather than participating in various activities in parks, visits to neighborhood parks are primarily for the purpose of going out, thereby functioning as adaptive spaces for urban residents by both satisfying people’s desire to go out and relieving the stress caused by the government’s restrictions on face-to-face activities. More specifically, by improving park services such as increasing community functions, upgrading walking trails, and expanding exercise amenities, neighborhoods can further reduce the spread of the virus, leading to a safer society and allowing parks to become hubs to improve the resilience of social-ecological systems against COVID-19.

South Korea gave an example of an effective initial response to COVID-19, but it was a short-term response that only focused on the care and prevention of infection, and therefore had the unfortunate effect of increasing social depression. However, to rebuild the collapsed social-ecological ecosystem beyond a simple response to the pandemic, it is necessary to improve resilience from a long-term perspective. Establishing a supportive environment that can absorb and reflect relevant impacts is also an important challenge in helping people recover from pandemics. Governments must create resilient environments such as parks that can absorb impacts and increase social adaptability. This is needed to expand the park infrastructure so it can function as a central point for the recovery of the social ecological system and to solve the usage issue of the park through landscape design. This study can encourage landscaping planning and management research that can respond to pandemics. In addition, the strategies learned during the COVID-19 era should be introduced into public space design as even if the transmission rate of COVID-19 has decreased and the system has been switched to living with the virus, the infected and people around them will still experience stress from it. Moreover, as the private realm and social distance are becoming morally important, not just for the epidemic but also for other socioenvironmental disturbances, a demand for resilient spaces will continuously grow. For that, the neighborhood parks will keep functioning as adaptive spaces that can continuously attract people and help them recover from stress.

## Figures and Tables

**Figure 1 ijerph-20-04259-f001:**
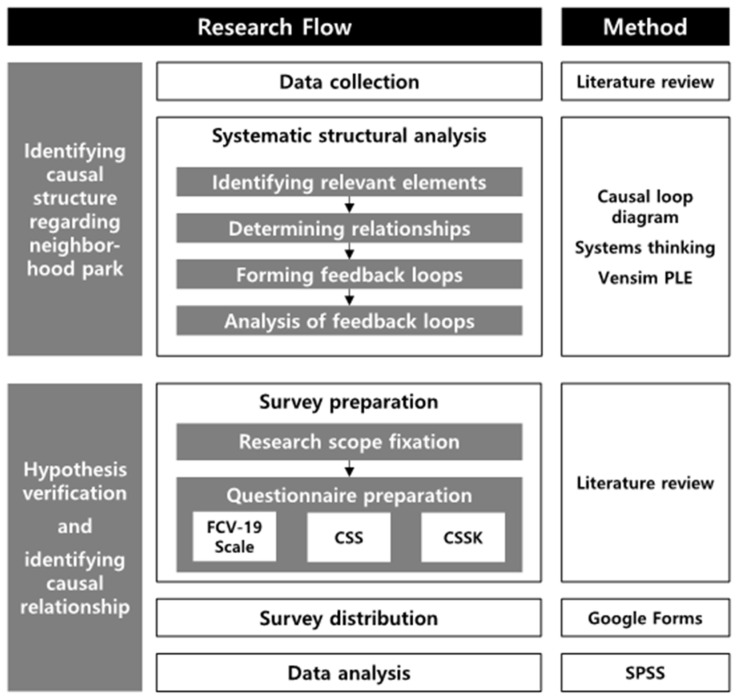
Research Framework.

**Figure 2 ijerph-20-04259-f002:**
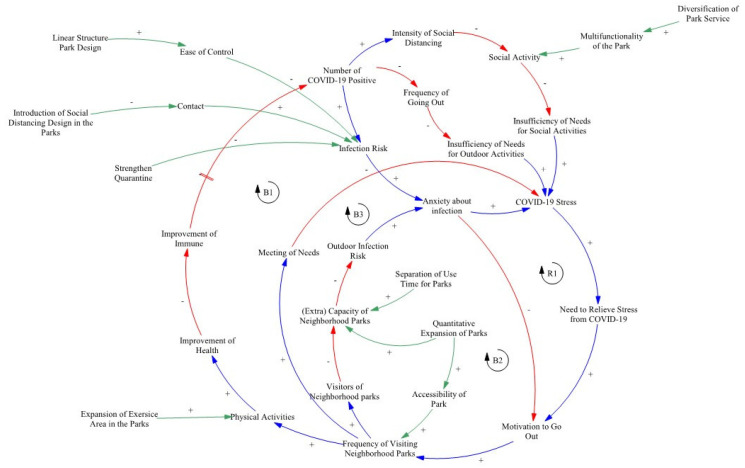
A causal loop diagram for COVID-19 around neighborhood parks.

**Figure 3 ijerph-20-04259-f003:**
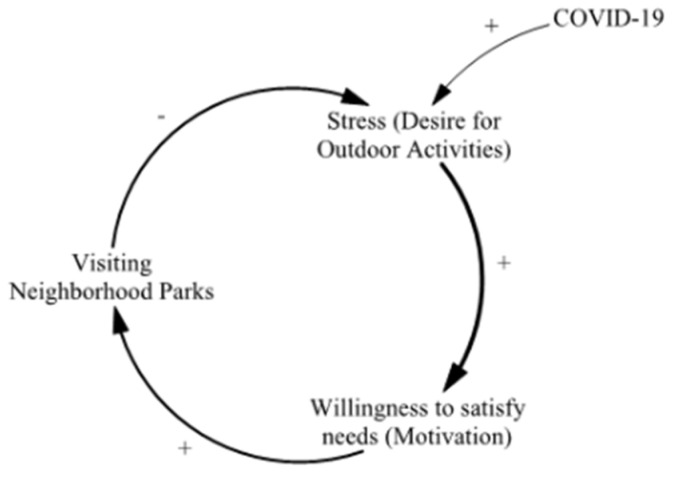
Simple structure of main variables.

**Figure 4 ijerph-20-04259-f004:**
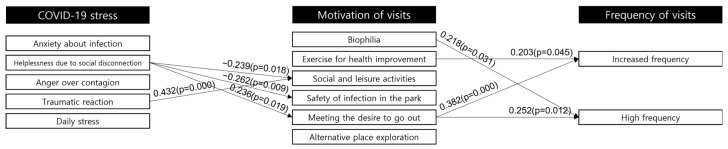
Causal relationships between stress, motivation for, and frequency of park visits.

**Table 1 ijerph-20-04259-t001:** Scale and references for questionnaire.

Classification	Scale	Reference
COVID-19 Stress	Fear	[27,29,30]
	Lethargy	[29]
	Anger	[29]
	Traumatic reaction	[30]
Motivation of visit	Physical activity	[31,32]
	Natural elements	[31,33]
	Stress relief	[31,32,33]
	Mental activity	[31,32,33]
	Social connection	[31,33]
	Related to COVID-19	[32,34,35,36]
Frequency of visits	.	[37]

**Table 2 ijerph-20-04259-t002:** Neighborhood park COVID-19 response feedback structure.

Feedback Loop	Property	Contents
B1	Balancing	Stress arises when the desire to engage in various activities is not satisfied due to the spread of COVID-19. To solve this problem, motivation to go out occurs, which leads to a visit to the neighborhood park. When needs are satisfied through a visit to the park, COVID-19 stress is reduced, forming feedback in which stress is balanced.
B2	Balancing	As the number of COVID cases increases, people worry more about infection and are more willing to stay inside rather than go out and engage in contact with others. This may lead people to stay home and thus improve the pandemic conditions; however, it does not initially improve COVID-19 stress.
B3	Balancing	Individuals visit neighborhood parks to relieve stress. By performing physical activities in the park, their health is improved and their immunity to COVID-19 increases. In the long term, this serves to prevent infection, forming a balancing loop in which the number of positive COVID-19 cases decreases.
R1	Reinforcing	If many people become concentrated in a neighborhood park of limited capacity due to the motivation to go out stemming from COVID-19 stress, then the risk of outdoor infection increases. This increases concerns about infection, forming reinforcing feedback that increases COVID-19 stress.

**Table 3 ijerph-20-04259-t003:** Reference of major relationship.

Variable	Polarity	Variable	Reference
Number of COVID-19 Positive cases	+	COVID-19 Stress	[28,39]
COVID-19 Stress	+	Motivation to go out	[40]
Frequency of Visiting Neighborhood Parks	+	Meeting of Needs	[41]
Frequency of Visiting Neighborhood Parks	+	Physical Activities	[42]
Capacity of Neighborhood Parks	−	Outdoor Infection Risk	[43]

**Table 4 ijerph-20-04259-t004:** General characteristics of respondents.

Characteristics	Variables	Frequency	Ratio
Sex	Male	23	23.5
	Female	75	76.5
Age	10s	6	6.1
	20s	49	50.0
	30s	13	13.3
	40s	18	18.4
	50s	9	9.2
	Over 60	3	3.1
Distance to neighborhood park (on foot)	Within 5 min	34	34.7
	Within 10 min	38	38.8
	Within 30 min	19	19.4
	Over 30 min	7	7.1

**Table 5 ijerph-20-04259-t005:** Principal component analysis of COVID-19 stress.

Division			Communality	Factor Loading	M (SD)	EV
Area		Item				
Factor 1	Anxiety about infection (α = 0.885)	I am worried about getting infected with COVID-19.	0.804	0.857	5.14 (1.506)	6.889
		I am worried about when and where I will be infected with COVID-19.	0.755	0.821	4.98 (1.586)	
		I’m worried that I might get infected with COVID-19 if I touch a handle in a public place.	0.620	0.756	4.57 (1.693)	
		I’m worried about getting infected with COVID-19 because of people around me.	0.656	0.740	5.03 (1.502)	
		I am worried about getting infected with COVID-19 in an enclosed place that I use frequently (e.g., elevators, public transportation).	0.631	0.690	4.80 (1.705)	
Factor 2	Helplessness due to social disconnection (α = 0.849)	I am depressed because I cannot do hobbies or cultural activities as I did before because of COVID-19.	0.726	0.827	4.88 (1.778)	2.404
		As social distancing continues for a long time, I feel disconnected from society.	0.725	0.816	4.19 (1.919)	
		Due to COVID-19, more time at home has lowered my will to live and made me lethargic.	0.752	0.722	4.17 (1.969)	
		It is hard to see family and friends very often because of COVID-19.	0.656	0.699	4.83 (1.811)	
Factor 3	Anger over contagion (α = 0.779)	I get angry when I see people going to high-risk facilities (e.g., pubs, clubs) where there is a risk of spreading COVID-19.	0.772	0.854	5.37 (1.778)	2.161
		I get angry at my boss, seniors, and adults in my family for forcing me to a dinner or meeting without considering the possibility of COVID-19 transmission.	0.646	0.768	5.13 (1.848)	
		I follow the quarantine rules well, but I get angry when other people do not follow them properly.	0.674	0.747	5.17 (1.687)	
		I get angry at religious people who insist on engaging in contact activities.	0.559	0.602	5.81 (1.564)	
Factor 4	Traumatic reaction (α = 0.756)	When I think of COVID-19, I sweat or my heart beats quickly.	0.767	0.851	1.40 (0.770)	1.762
		I have been contemplating suicide because of COVID-19 stress.	0.718	0.838	1.18 (0.615)	
		It is hard to concentrate because I’m worried about COVID-19.	0.703	0.725	1.83 (1.149)	
		It is difficult to get enough sleep due to the psychological pressure of COVID-19.	0.504	0.610	1.96 (1.235)	
Factor 5	Daily stress (α = 0.797)	Daily life has become terrifying due to COVID-19.	0.771	0.703	4.17 (1.675)	1.118
		When COVID-19 news is updated on TV or social media, it is stressful.	0.689	0.700	4.31 (1.796)	

**Table 6 ijerph-20-04259-t006:** Principal Component Analysis of Motivation to Visit Neighborhood Parks.

Division			Communality	Factor Loading	M (SD)	EV
Area		Item				
Factor 1	Biophilia (α = 0.879)	To enjoy the beautiful scenery	0.827	0.840	4.41 (1.931)	4.012
		To rest	0.722	0.835	5.21 (1.778)	
		To experience nature	0.756	0.811	4.47 (1.933)	
		For psychological stability and relaxation	0.717	0.751	5.26 (1.575)	
		To energize	0.642	0.655	5.49 (1.372)	
		A visit to the park relieves tension	0.572	0.593	4.03 (2.023)	
Factor 2	Exercise for health improvement (α = 0.645)	Because you can go for a walk or jog	0.735	0.744	5.84 (1.249)	2.013
		To exercise in the park	0.643	0.731	4.54 (1.742)	
		Because you can breathe the fresh air	0.603	0.514	5.18 (1.719)	
		To prevent disease through healthy living	0.677	0.484	3.67 (2.014)	
Factor 3	Social and leisure activities (α = 0.616)	To meet new people	0.662	0.804	1.73 (1.273)	1.952
		To read a book in the park	0.598	0.708	1.95 (1.424)	
Factor 4	Safety from infection in the park (α = 0.699)	Because the park is safer from COVID-19 than other places	0.826	0.892	4.45 (1.873)	1.911
		Because the park is sanitary	0.671	0.658	3.30 (1.682)	
Factor 5	Meeting the desire to go out (α = 0.797)	To satisfy the desire to go out	0.816	0.873	4.44 (1.959)	1.837
		To get out of the house	0.790	0.827	4.51 (1.922)	
Factor 6	Alternative place exploration (α = 0.623)	To meet friends	0.724	0.773	3.50 (2.112)	1.584
		No limit on the number of people	0.736	0.645	3.41 (2.004)	
		More free time because daily life has shifted to working at home or online	0.591	0.509	3.14 (1.948)	

## Data Availability

Data sharing is not applicable.

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
