# Peer review of "The Perceptions and Use of Urban Neighborhood Parks Since the Outbreak of COVID-19: A Case Study in South Korea"

_ijerph, 2023, doi:10.3390/ijerph20054259_

Round 1

Reviewer 1 Report

In general, this paper deals with an important but mostly unexplored issue about park use behaviours by causal loops under the background of COVID-19 in South Korea. The findings are practical for urban policymakers and park planners. The conclusions are also helpful to adapt to the impact of the pandemic in the era of coexisting with COVID-19. However, I still have some concerns about the research design and conduct after reading the full manuscript. Please see my detailed comments as follows.

1. In section 2.1.1, line 140, the author presents the method to identify relevant elements of casual loops by searching articles in databases. However, it is not clear the criterion of journal articles to be selected in the study. For example, how many results were retrieved? What kinds of articles are involved in this research and what kinds are excluded? Is the study of these articles conducted in South Korea, or in the context of other parts of the world?

2. In section 2.2.1, line 181, there are 114 responses and only 98 of them are valid (line 206). The number of samples is quite limited if the author has done the survey by random sampling across the whole country. The author has claimed that it is sufficient for the analysis (section 3.2.1, line 255), but the travel behaviour to parks is highly correlated to the built environment, and the heterogeneity of the built environment across the country is huge. Therefore, the normal distribution of 30 samples proposed by the author is less convincing and the samples in this study may not cover this heterogeneity. If the survey was done in a specific place or city, then the author should describe it in a proper way.

3. As a case study in South Korea, more discussions should be done by comparing the results with previous works in section 4. For example, what are the specific findings in the context of South Korea? What is the difference between the specific findings in this study and the results from other countries? Do the new findings support or contradict the specific recognition in urban planning or management in South Korea?

4. As a minor aspect, the original questionnaire designed in the survey should be submitted as supplementary materials to support the review process.

Reviewer 2 Report

I found this paper only of minor interest and relevance, because the research question are basically very simple ("is the frequency of park visits positively correlated to COVID-19 induced stress levels?") and the outcome rather predictable. The causal loop diagram as such is well constructed, but fails to include spatial factors, such as distance to the neighborhood park or availability of private gardens (this would be an important factor in many North-American and European countries). The cultural and spatial specificity of the South-Korean situation is thus not addressed.  

It was not clear to me which set of neighborhood parks was included in the study (only in Seoul, or also elsewhere?). 

Some sentences were wholly incomprehensible to me (e.g. line 65-66; line 168-170; line 257-259) or seemed to state the obvious coded in a jargon-like language (line 69-73; line 88-92).

I do not see how the findings of this study can help landscape planning and management - apart from the rather obvious observation that neighborhood parks are an asset for cities and their inhabitants. In how far these parks are 'resilient spaces' (line 441) merits a fundamental discussion, but I am afraid that this study does not contribute a whole lot to that discussion. 

Reviewer 3 Report

The article is very well structure. 

Some detailed comments:

1.       Line 276: Define KMO properly.

2.       Table 5: Define EV properly in the table

3.       Discussion: I would recommend to divide the discussion into subsections to highlight the findings for each hypothesis mentioned in the introduction.

Reviewer 4 Report

1. The article covers a very important aspect of an impact that urban parks have on physical and mental well-being, especially during a pandemic. The authors highlighted that people’s physical and mental health is related to their biophysical environment, so it is important to focus on and manage this socioecological system. Even though there are lots of research done on this topic, it is still essential to provide an indepth insight into the interaction of the psychological, social, and ecological variables. Moreover, the novelty of the article is based on the integrated and systematic investigation how the relationship between the perceptions and use of urban neighborhood parks is affected by the COVID-19 outbreak (in South Korea). The presented study not only reveals the increased visits to neighborhood parks but also aims at explaining the reasons why people visit parks during the pandemic. 

2. It would be essential to provide a reference for "Corona blue" term (line 51-52).

3. I would argue with the statement that "the external conditions other than the behaviors in the parks was assumed as similar to the early stage of virus outbreak" (lines 129-130). The early stage of virus outbreak brought a lot of reactions that changed a socio-psychological profile of the whole societies, safety and law regulations included.

4. For the population of over 50 mln people in South Korea (2021), the sample of 114 responses seems highly insufficient, also because the survey was produced in Google Forms and distributed online. Over 76% of the respondents were women, which also do not provide a representative sample. Maybe if the sample was bigger, other more advanced analysis was possible (not only correlations).

5. I understood that a causal loop diagram for COVID-19 around neighborhood park presented in Figure 2 is based on literature review, which may be a very subjective analysis of both positive and negative feedback. maybe it would be worth explaining on what basis the specific details of the diagram were introduced and link them to the literature (source of research that focused on that specific variable). The figure gives an outlook of the interactions, but some points are not as obvious as commented on the graph, eg. need to relieve stress from COVID-19 does not necessarily lead to the increased motivation to go out when a person is aware of the infection risk. The study leaves behind the notion of online social interactions that in many cases compensated for the possibility of face-to-face interactions. Nevertheless, the causal loop diagram might be an interesting alternative for the graphic presentation of the interactions between the variables.

6. It is worth to discuss the general process of environmental psychology in which cognition leads to perception and then behavior. Since the change in behavior and perception often leads to the change in cognition, it is necessary to broaden the perspective of the analysis of the interaction between behavior, perception, and cognition. In this context, the beginning of the Discussion section should be also modified (lines 334-338). 

Round 2

Reviewer 1 Report

The author has responded to my concerns and their amendment has significantly improved the paper.